# Different patterns of pneumothorax in patients with soft tissue tumors treated with pazopanib: A case series analysis

**Hisaki Aiba**[1]*, **Hiroaki Kimura**[1], **Satoshi Yamada**[1], **Hideki Okamoto**[1], **Katsuhiro Hayashi**[1,2], **Shinji Miwa**[1,2], **Yohei Kawaguchi**[1], **Shiro Saito**[1,2], **Takao Sakai**[1], **Tsutomu Tatematsu**[3], **Ryoichi Nakanishi**[3], **Hideki Murakami**[1]

**1** Department of Orthopaedic Surgery, Nagoya City University, Nagoya, Japan, **2** Department of Orthopaedic Surgery, Kanazawa University Graduate School of Medical Sciences, Kanazawa, Japan, **3** Department of Oncology, Immunology and Surgery, Nagoya City University, Nagoya, Japan

\* hisakiaiba@yahoo.co.jp

**Data Availability Statement:** All relevant data are within the paper and its Supporting Information files.

## Abstract

To investigate pneumothorax patterns in pazopanib treatment by focusing on the positional relationship between the visceral pleura and metastatic lung tumor, we examined 20 patients with advanced soft tissue tumors who developed lung metastases and underwent pazopanib treatment between 2012 and 2019. Pneumothorax was classified into two types based on the location of the metastatic lesion around the visceral pleural area before pazopanib treatment: subpleural type, within 5 mm from the pleura; and central type, >5 mm from the pleura. We investigated the rates of pneumothorax and the associated risk factors. Five patients experienced pneumothorax (three subpleural and two central types). Cavitation preceded pneumothorax in 83% of patients and led to connection of the cavitated cyst of the metastatic lesion to the chest cavity in the shorter term in patients with the subpleural type. Conversely, a more gradual increase in the cavity size and sudden cyst rupture were observed in the central type. The risk factors for pneumothorax were cavitation after initiating pazopanib and intervention before pazopanib, either ablation or surgery. The location of the metastatic lesions was not a risk factor for the occurrence of pneumothorax. In conclusion, pneumothorax is an adverse event associated with pazopanib treatment. Therefore, attention must be paid to predisposing factors such as the formation of cavitation after pazopanib initiation and previous interventions to the lungs. Moreover, because subpleural pneumothorax tends to occur earlier than the central type, a different time course can be anticipated based on the positional relationships of the metastatic lesions to the visceral pleura.

## Introduction

Pazopanib, a multi-channel kinase inhibitor, was approved for the treatment of soft tissue tumors in 2012 based on the results of a phase III clinical trial. Although pneumothorax was

**Funding:** The author(s) received no specific funding for this work.

**Competing interests:** The authors have declared that no competing interests exist.

not a common adverse event in this trial, with an incidence of only 3.3% [1], subsequent real-world studies indicated a higher incidence of 7%–15% [2, 3]. Because this critical adverse event interrupts treatment and may even require treatment cessation [2], management of pneumothorax is essential in the clinical setting. Based on our clinical experience, we identified two types of pneumothorax in patients treated with pazopanib from two different metastatic lesions: subpleural and central types (Figs 1 and 2). This study aimed to investigate the pattern of pneumothorax. We proposed different types of pneumothorax focusing on the positional relationship between metastatic lesions and the pleura and presented their clinical features. We also proposed an appropriate treatment strategy for pneumothorax development after pazopanib treatment.

## Materials and methods

### Patients and pazopanib protocol

From 2014 to 2019, 32 patients with advanced soft tissue tumors were treated with pazopanib at the Nagoya City University Hospital. All patients received at least one cycle of an anthracycline-containing regimen. All patients had a performance status (PS) of 0–2 before pazopanib treatment, and the inclusion criteria in the package insert [4] were followed. None of these patients had a past medical history of cavity lung lesions, infection (e.g., tuberculosis, non-tuberculous mycobacterial infection, aspergilloma), sarcoidosis, or primary lung cancer. After excluding 12 patients with extra-pulmonary metastases, a total of 20 patients were included in the analysis. The patients' baseline characteristics, including age, sex, PS, pathological diagnosis, number of lung lesions, and pretreatment history of lung disease, were retrospectively reviewed from the medical records following full anonymization. The initial pazopanib dose was 800 mg (once daily, orally), and the dose was reduced based on the occurrence of adverse events (e.g., diarrhea, nausea, loss of appetite, hypertension, hand-foot syndrome) or patient status.

### Radiographic analysis and definition of pneumothorax

Chest computed tomography (CT, 2-mm slice) was performed before the initiation of pazopanib and repeated at every visit for 3–6 months to monitor metastatic lesions. Additional CT was performed in accordance with the patient's symptoms. All metastatic target lesions were evaluated by a single observer (HA) who classified the target lesions as subpleural (i.e., located within 5 mm or attached to the visceral pleura) or central (i.e., located > 5 mm from the visceral pleura) based on the images before the initiation of pazopanib treatment. Measurements were evaluated at the point where the distance between the visceral pleura and the tumor ridge was tangentially minimized in axial CT images. In addition, cavitation, defined as a degenerated change of metastatic lesion into a cyst storing air after pazopanib [5], was carefully observed as it preceded the pneumothorax.

### Treatment of pneumothorax

The severity of pneumothorax was categorized based on the Common Terminology Criteria for Adverse Events CTCAE 5.0 and the size of the collapse. The size was evaluated by measuring the length from the lung margin to the chest wall on chest radiography [6]. "Small" pneumothorax was defined as "small rim of air around the lung (< 2 cm)," "moderate" as "collapsed about halfway to the heart," and "complete" as "airless lung or separation from the diaphragm." Treatment was performed according to the guidelines of the American College of Chest Physicians or British Thoracic Society [6]. Asymptomatic patients or those with a small pneumothorax were closely monitored and/or administered oxygen therapy. Further, pazopanib

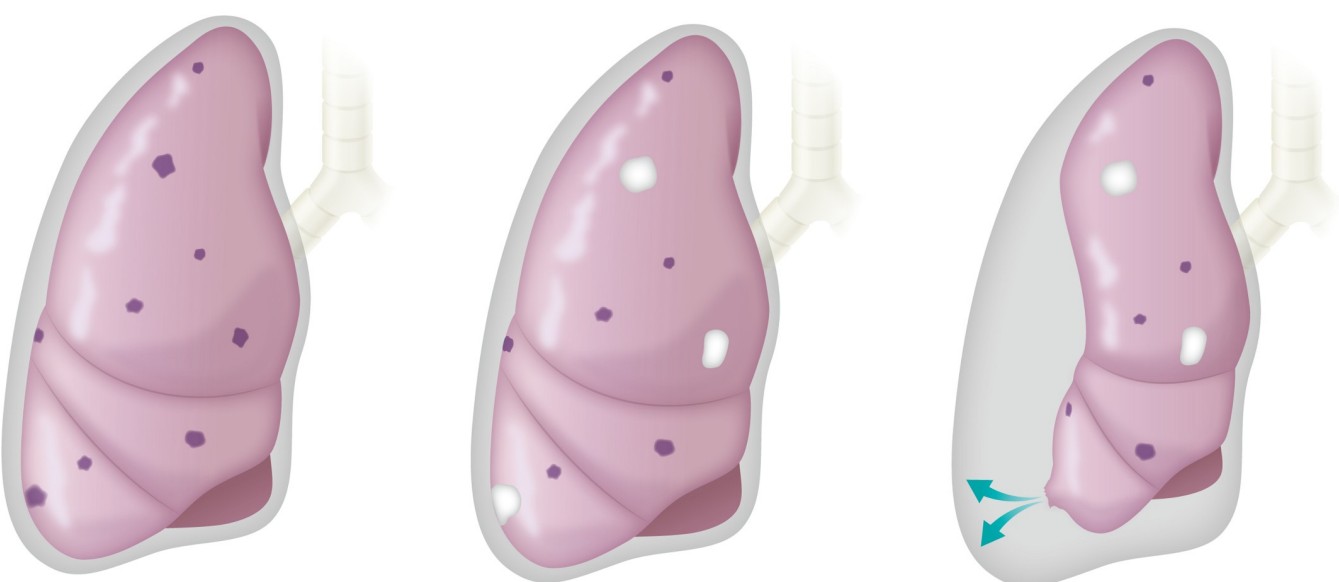

**Fig 1. Subpleural type pneumothorax.** Panels a-c indicate the scheme of the subpleural type pneumothorax. The metastatic tumors are indicated as purple lesions (a). After initiating pazopanib, the tumors shrank and were sometimes accompanied with the formation of a degenerated cavity when pazopanib was effective on the tumors (b). Regarding the metastatic lesion located around or attached to the visceral pleura, pneumothorax (blue arrow) occurred at the junction between the chest cavity and ruptured cavity (c).

administration was discontinued for 3–4 weeks until recovery of pneumothorax. Meanwhile, symptomatic patients or those with moderately sized to complete pneumothorax (> 2 cm) were primarily treated with drainage or aspiration using a one-way valve or water seal device for 5–7 days. Meanwhile, pleurodesis with 5–10 KE of picibanil (Chugai Pharmaceutical, Tokyo, Japan) was considered for repeated or refractory pneumothorax. Video-associated thoracoscopic surgery (VATS) was performed only for the removal of resilient blebs/bullae or fistulas.

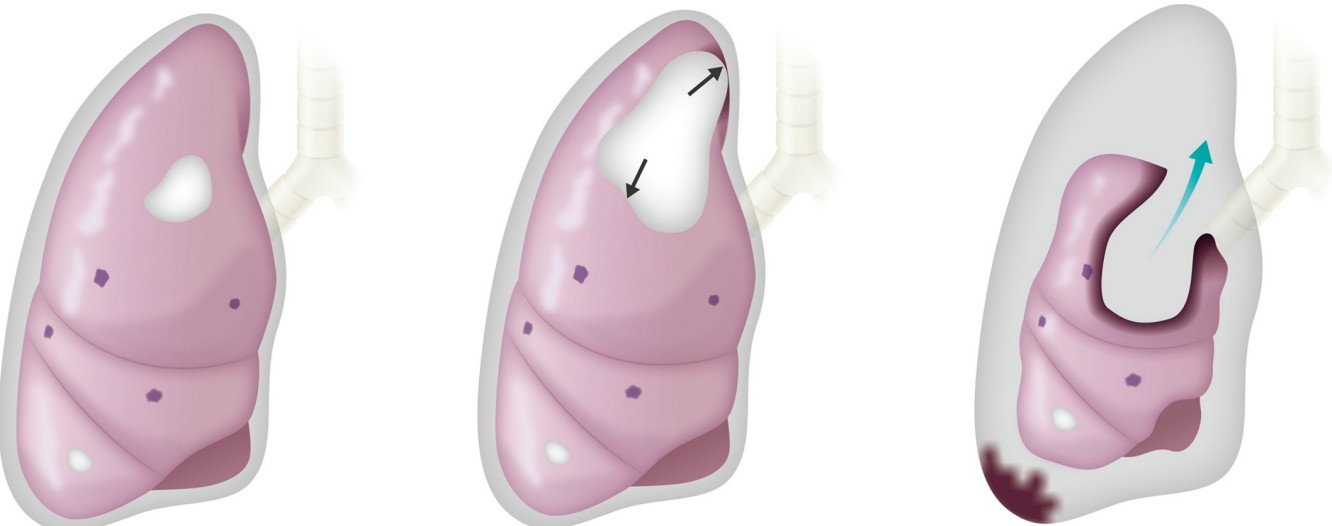

**Fig 2. Central type pneumothorax.** Panels a–c indicate the scheme of the central type pneumothorax. After initiating pazopanib, some metastatic lesions changed to degenerated cysts. After cavitation, air leaked (black arrows) continuously to the cavity (a and b). Ultimately, the high pressure in the cavity caused the rupture of the cyst and pneumothorax (blue arrow, c). This pattern sometimes forms an intractable large fistula in the chest cavity and is accompanied by pleural dissemination of tumors and massive bleeding.

### Statistical analysis

All values are presented as either mean ± standard deviation or median with range, depending on the distribution. The chi-square test was used to compare the categorical variables. All statistical analyses were conducted using SPSS version 24 (IBM, Chicago, IL, USA), and P < 0.05 was considered significant.

### Ethical approval

The study was approved by the Ethics Committee of Nagoya City University (No 60-19-0029) and was conducted in compliance with the guidelines of the 1975 Declaration of Helsinki. Due to retrospective setting, the ethical committee permitted exemption to acquire written consents to use their medical records/samples and to participate in the study. Instead, the protocol of this study was announced in the form of opt-out on the web site in Clinical Research Management Center of Nagoya City University. Those who were not willing to participate were excluded.

## Results

The mean patient age was 56.3 ±18.8 years, and 14 patients were men and 6 were women (Table 1). The histological subtypes included undifferentiated pleomorphic sarcoma in 9 patients, leiomyosarcoma in 3 patients, and others in 8 patients. Multiple metastatic lesions (> 10 lesions) were observed in 10 patients. The median pazopanib treatment period was 3.5 months (range, 1–51 months). At the end of the study, 16 patients died of the disease, while four patients were alive with the disease. The details of the patients' characteristics are shown in S1 Table.

### Incidence of pneumothorax

Five patients were diagnosed with pneumothorax. The percentage was 15.6% (5/32) considering all patients and 25.0% (5/20) considering patients with lung metastasis. The symptoms of the first pneumothorax were dyspnea (n = 3; on exertion, 1; and resting condition, 2) and

**Table 1. Analysis of the different characteristics affecting the incidence of pneumothorax (chi-square test).**

| Characteristics | | Incidence of pneumothorax | p-value |
|---|---|---|---|
| Sex | Male | 3/14 | 0.848 |
| | Female | 2/6 | |
| Histology | UPS | 2/8 | 1.0 |
| | Others | 3/12 | |
| Previous interventions to lungs before pazopanib administration | Yes | 3/4 | 0.010 |
| | No | 2/16 | |
| Cavitation of the metastatic lesion | Yes | 5/6 | <0.001 |
| | No | 0/14 | |
| Treatment response to pazopanib | PR | 4/9 | 0.069 |
| | SD or PD | 1/11 | |
| Presence of subpleural lesions | Yes | 4/18 | 0.389 |
| | No | 1/2 | |
| Initial dose of pazopanib (mg/day) | 800 | 4/7 | 0.015 |
| | Under 800 | 1/13 | |

Abbreviations: UPS, undifferentiated pleomorphic sarcoma; PR, partial response; SD, stable disease.

chest pain (n = 1); one patient had no complaints. The median time from the initiation of pazopanib to the first pneumothorax was 3 months (range, 2 weeks to 20 months). In total, three and two patients developed pneumothorax from the subpleural and central lesions, respectively. The first pneumothorax occurred at 2 weeks, 3 weeks, and 8 months after initiation of pazopanib in the three patients with the subpleural type and at 3 months and 20 months in the two patients with the central type. Representative cases are shown in Figs 3 and 4. Pazopanib treatment was restarted in 3 of the 5 patients after recovery from pneumothorax only if there were no other drug choices, but pneumothorax recurred in all three patients. Overall, the total incidence of pneumothorax was 15 times in five patients.

With respect to the location of metastatic lung lesions, 285 lesions per patient (14.3 ±12.9) were detected before the induction of pazopanib. Seventeen patients had both central and subpleural lesions, two patients had only central lesions, and one patient had only subpleural lesions. Of these, 171 lesions (8.6 ±9.3 lesions per patient) were subpleural, and 114 lesions (5.7 [±4.5] lesions per patient) were central. Twelve cases of pneumothorax, including recurrence, were observed in subpleural lesions (7.0%, 12/171); three cases of pneumothorax (2.6%, 3/114) were from central lesions. There was no statistically significant difference in the incidence of pneumothorax according to the tumor location (chi-square, p = 0.104).

## Treatment

For the subpleural type, asymptomatic patients or those with small size pneumothorax (2 incidents in 2 patients) were administered conservative treatment, while a chest drainage tube was inserted in patients with moderately sized or complete pneumothorax (10 incidents in 3 patients). In total, the treatment efficacy of chest tube insertion was 60% (6/10). For resistant cases, pleurodesis was performed. This resulted in remission in all cases.

All cases of central type pneumothorax involved a complete pneumothorax (three incidents in two patients). The pneumothorax was uncontrollable in all cases using a chest tube; thus, pleurodesis or VATS was performed, depending on the recovery after re-expansion of the lung and the estimated size of the fistula.

## Factors associated with pneumothorax

Cavitation was observed after the initiation of pazopanib in 30% (6/20) of patients and preceding the first pneumothorax in 83% of the patients (5/6); however, none of the patients without pneumothorax also developed cavitation (p<0.001, chi-square analysis). With respect to classification according to treatment response, 45% (9/20) and 55% (11/20) were evaluated as partial responders and inadequate responders (stable disease or progressive disease). Cavitation of the lung lesion was observed in 50% (5/10) of the partial responders and 10% (1/10) of the inadequate responders (stable disease or progressive disease), but there was no significant association between treatment response to pazopanib and cavitation (p = 0.051). Likewise, pneumothorax occurred in 44% (4/9) of partial responders and 9% (1/11) of inadequate responders (stable disease or progressive disease), but there was no significant association between treatment response to pazopanib and cavitation (p = 0.069). Moreover, the predisposing factor for pneumothorax was previous lung treatment (e.g., metastasectomy or radiofrequency ablation for the resection of metastatic lesions) before pazopanib administration (75% [3/4] vs. 13% [2/16], p = 0.01, Table 1), although the location of the pneumothorax was not directly related to the surgical site. In addition, the initial dose of pazopanib was associated with the rate of pneumothorax (57% [4/7] vs. 8% [1/13], p = 0.015, Table 1) in comparison to the maximum dose (800 mg).

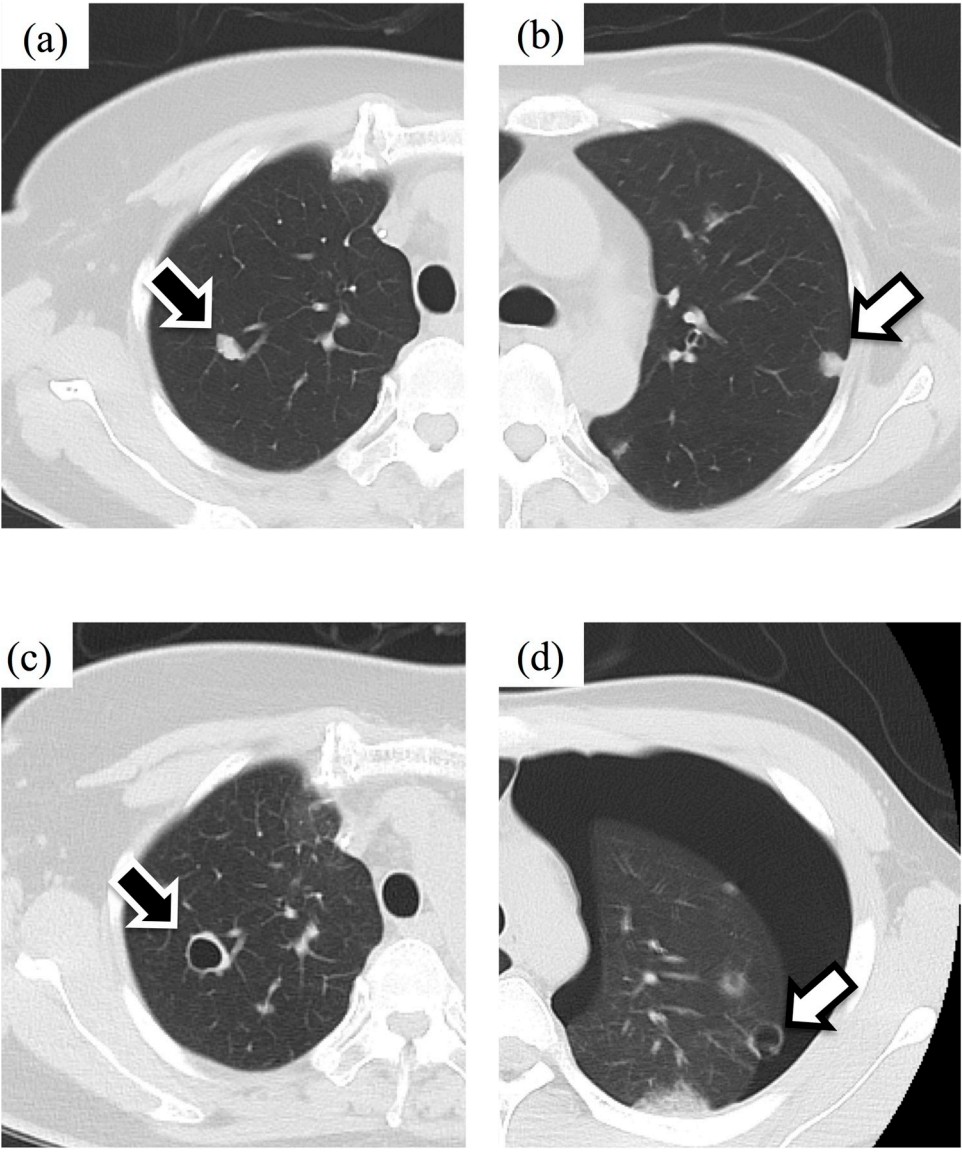

**Fig 3. Representative case of pneumothorax from a subpleural type lesion.** A 57-year-old woman was diagnosed with unclassified pleomorphic sarcoma in the right lower leg with multiple lung metastases. Before initiating pazopanib (800 mg), both central (black arrow) and subpleural type (white arrow) metastases without cavitation were observed (a and b). Cavitation of the metastatic lesion was observed in the lung parenchyma (central type, black arrow, c) and around the subpleural lesion after 2 weeks. One month after initiating pazopanib, pneumothorax occurred from the subpleural type lesion (white arrow, d), but the central lesion did not rupture. Because of the good response of the multiple lung metastases to pazopanib, the patient was restarted on pazopanib after the first pneumothorax. Bilateral pneumothorax from the sub-pleural areas (left [the same place as first metastasis], right [S5], left [S5], left [S1+2]) occurred consecutively. As regards cavitation in the central area, the pre-existing cavitation had slightly increased in size but never ruptured.

## Discussion

Pneumothorax is a serious adverse event of pazopanib treatment, and there have been case reports and series on pneumothorax occurring after pazopanib treatment [2, 7, 8]. In Japan, Nakano et al. reported a 10.3% incidence of pneumothorax, and the risk factor was lung metastatic tumor measuring > 3 cm and a history of pneumothorax [2]. Although the cause was

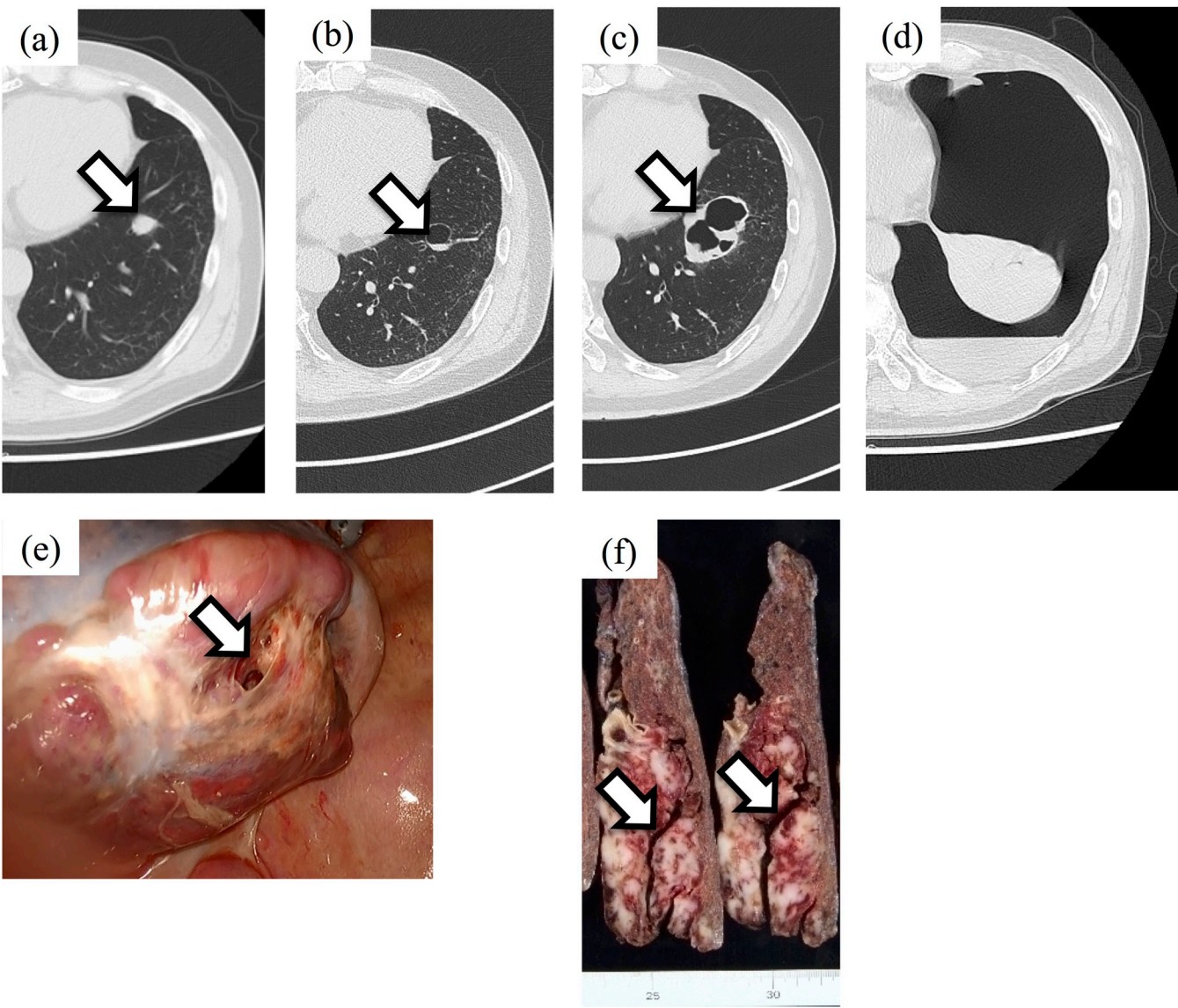

**Fig 4. Representative case of pneumothorax from a central type lesion.** A 63-year-old man was diagnosed with unclassified pleomorphic sarcoma in the right thigh without any metastasis. The patient developed a single metastasis after 1 year; thus, lobectomy with video-associated thoracoscopic surgery was performed. Another single metastasis of the left lobe (central lesion, white arrow, a) was observed; thus, pazopanib 600 mg was initiated. (b) After 2 months, cavitation of the metastatic lesion was observed (white arrow). (c) The cavitation increased gradually in size, and (d) after 6 months, the patient developed dyspnea and was then diagnosed with severe pneumothorax with a mediastinal shift and accompanying bleeding. (e) Chest drainage was performed; however, the patient's status did not improve. Thus, lobectomy with intractable fistula (white arrow) was performed. (f) Macroscopic image of the resected specimen indicating a massive necrotic area (white arrow) along the fistula to the degenerative cavity.

unclear, the incidence rate was higher than that in the multi-center clinical trials of pazopanib (3% in the PALETTE study) [1]. In this study, we classified pneumothorax into two types: subpleural and central. In the subpleural type, the metastatic lesion was located around or attached to the visceral pleura. Typically, after cavitation of the metastatic tumor, pneumothorax occurs at the junction between the chest cavity and the ruptured cavity. In contrast, in the central type, the metastatic lesion was located distal to the pleura. After cavitation, the check-valve mechanism occurred, or air leaked continuously into the cavity. Finally, the high pressure

caused enlargement of the cavity, resulting in rupture and pneumothorax. This indicates that careful monitoring is essential to note a gradual increase in the cavity size.

We observed that pneumothorax from subpleural lesions occurred much faster after the initiation of pazopanib and was occasionally refractory when compared with that from central lesions. Meanwhile, central pneumothorax is slower in onset but is sometimes intractable owing to the formation of an untreatable large fistula between the visceral pleura and the cavity (caused by continuous massive air leakage into the cavity). Despite the limited number of cases in this study, this classification may be useful for clinicians to predict the timing and severity of pneumothorax.

We assumed that the cause of the central pneumothorax was similar to pneumatocele. Pneumatocele is defined an air-filled cyst that develops within the lung parenchyma. It often occurs in respiratory infections, trauma, or mechanical ventilation [9, 10]. Although the exact mechanism is unclear, it has been considered to be caused by parenchymal necrosis and check-valve bronchiolar obstruction [11]. Another study proposed local collection of air in the interstitial tissue [12]. In our case series, a gradually enlarging cyst was observed, and a necrotic tract from the cyst to the pleural membrane was observed on the resected specimen (Fig 4). This finding suggested that, compared with subpleural pneumothorax, central type pneumothorax is preceded by more extensive necrosis.

The formation of a cavity is often facilitated by several antiangiogenic agents, including pazopanib [13, 14]. In this study, cavitation of the tumor preceded pneumothorax in most cases. Cavitation of lung metastases may also be associated with the treatment response to pazopanib and, simultaneously, with the risk of pneumothorax [15, 16], thus making treatment challenging. In our department, continuation or re-administration of pazopanib is only considered when pazopanib is effective at controlling tumor progression or when treatment options are limited. However, all patients developed recurrence of pneumothorax; thus, this should be carefully considered in patients with a history of pneumothorax after pazopanib treatment, and other alternative chemotherapeutics should also be considered. A balance between pazopanib efficacy and adverse events is important for planning treatment.

The exact reasons for the high rate of pneumothorax occurrence are unclear, but the selection of patients with naturally cyst-forming characteristics might have influenced the results of this study. Upon analyzing the natural characteristics of sarcomas, cyst-forming metastases in the lung, unrelated to pazopanib, had been previously reported [17]. In this study, we analyzed patients who had pre-existing cavitation (Pt No. 2) and cystic changes in metastatic lesions after cessation of pazopanib (Pt No. 8), which acted as the initiation point for the occurrence of pneumothorax. This phenomenon was supported by the fact that secondary pneumothorax has rarely been reported in cases of kidney cancer, for which pazopanib has also been approved [18].

This study has some limitations. First, during the early phase of the study period, new drugs for advanced soft tissue sarcoma, including eribulin or trabectedin, were not approved; thus, treatment options were limited. This may have increased the incidence of pneumothorax in high-risk patients (i.e., those with a previous history of pneumothorax or with a degenerative cavity) and recurrence of pneumothorax. Second, this study was performed at a single institution, and a limited number of patients were included in the study, thus posing a limitation toward performing multivariate analyses and reaching a precise conclusion. Moreover, only one race (East Asian) was included in this study. Although it has been proven that the metabolism of pazopanib does not differ among races [19], the dose of pazopanib (800 mg) used in the PALETTE study might have been excessive for the population in our study, as East Asians are characterized by relatively small body stature. In fact, a real-world study of pazopanib usage in Japan revealed that the average dose of pazopanib was approximately 600 mg [3]. The

relationship between pneumothorax and the dose of pazopanib was not proven, and the dose might have influenced the study findings. Third, detection bias for pneumothorax affected the results of this study. The symptoms of pneumothorax are highly varied. Some patients are nearly asymptomatic while others experience such symptoms as dry cough, chest pain, and dyspnea, resulting in severe distress during breathing. Efforts to identify pneumothorax might have influenced the results. However, to the best of our knowledge, this is the first study to classify pneumothorax after pazopanib treatment according to its pattern.

## Conclusion

Pneumothorax is considered a severe adverse event of pazopanib treatment. Although precise prediction might be difficult, close attention must be paid to the predisposing factors, including the formation of cavitation after the initiation of pazopanib and previous interventions performed on the lungs. Moreover, from the viewpoint of the positional relationship, where the subpleural type tends to occur earlier than central pneumothorax and because the central type may be challenging to treat owing to the larger fistula between the chest cavity and the degenerated cavity, different behaviors can be anticipated. Thus, the location of the metastatic lesion should be considered during decision-making in order to observe such lesions in the lungs.

## Supporting information

**S1 Table. Summary of patient characteristics and treatment courses.**
(DOCX)

## Acknowledgments

We thank the staff members of the Division of Pathology of Nagoya City University Hospital for evaluating the histological specimens and staff members of the Department of Radiology of Nagoya City University Hospital for evaluating the radiological images.

## Author Contributions

**Conceptualization:** Hisaki Aiba, Hiroaki Kimura, Satoshi Yamada, Ryoichi Nakanishi, Hideki Murakami.

**Data curation:** Hisaki Aiba, Hiroaki Kimura, Satoshi Yamada, Hideki Okamoto, Shinji Miwa, Yohei Kawaguchi, Tsutomu Tatematsu, Ryoichi Nakanishi, Hideki Murakami.

**Formal analysis:** Hisaki Aiba, Shinji Miwa, Yohei Kawaguchi, Shiro Saito, Hideki Murakami.

**Funding acquisition:** Hisaki Aiba, Hideki Murakami.

**Investigation:** Hisaki Aiba, Hiroaki Kimura, Satoshi Yamada, Katsuhiro Hayashi, Yohei Kawaguchi, Hideki Murakami.

**Methodology:** Hisaki Aiba, Hideki Okamoto, Katsuhiro Hayashi, Hideki Murakami.

**Project administration:** Hisaki Aiba, Hiroaki Kimura, Hideki Okamoto, Yohei Kawaguchi, Hideki Murakami.

**Resources:** Hisaki Aiba, Shinji Miwa, Hideki Murakami.

**Software:** Hisaki Aiba, Satoshi Yamada, Hideki Okamoto, Shinji Miwa, Hideki Murakami.

**Supervision:** Hisaki Aiba, Satoshi Yamada, Katsuhiro Hayashi, Shiro Saito, Takao Sakai, Tsutomu Tatematsu, Ryoichi Nakanishi, Hideki Murakami.

**Validation:** Hisaki Aiba, Satoshi Yamada, Shiro Saito, Takao Sakai, Hideki Murakami.

**Visualization:** Hisaki Aiba, Takao Sakai, Tsutomu Tatematsu, Hideki Murakami.

**Writing – original draft:** Hisaki Aiba, Shiro Saito, Hideki Murakami.

**Writing – review & editing:** Katsuhiro Hayashi, Shinji Miwa, Yohei Kawaguchi, Takao Sakai, Tsutomu Tatematsu, Ryoichi Nakanishi, Hideki Murakami.

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
