## [Decision Letter · Decision Letter 0]

27 May 2021

PONE-D-21-10938

Different patterns of pneumothorax in patients with soft tissue tumors treated with pazopanib: A case series analysis

PLOS ONE

Dear Dr. Aiba,

Thank you for submitting your manuscript to PLOS ONE. After careful consideration, we feel that it has merit but does not fully meet PLOS ONE’s publication criteria as it currently stands. Therefore, we invite you to submit a revised version of the manuscript that addresses the points raised during the review process.

We look forward to receiving your revised manuscript.

Kind regards,

Robert S Benjamin

Academic Editor

PLOS ONE

Additional Editor Comments:

This is an interesting study that presents important and useful information, but it needs some clarification. Please recheck the numbers. You state that there were 32 patients treated, 4 were excluded due to extrapulmonary metastases, but that left 19 patients. What about the other 9? The manuscript focuses on 20 patients. Where did the other 1 come from? More discussion should be placed on the extraordinarily high response rate, which is typically 5%, and whether the "nonresponding" patient who had pneumothorax had a response that did not qualify for PR. Are Japanese patients more likely to respond to this therapy or simply more likely to develop pneumothorax than caucasians. Please address these questions as well as others raised in the review process.

Journal Requirements:

2. Please include the full name of the ethics committee that approved your study in the manuscript Methods.

3. Please state whether the data were obtained retrospectively from medical records. If so, in the ethics statement in the manuscript and in the online submission form, please provide additional information about the patient records/samples used in your study, including: a) whether all data were fully anonymized before you accessed them; b) the date range (month and year) during which patients' medical records/samples were accessed; c) the source of the medical records/samples analyzed in this work (e.g. hospital, institution or medical center name).

In addition, we note that you obtained consents from participants to take part in your study. In the Ethics Statement on the online submission form and the manuscript Methods , please clarify the context in which consent was obtained, and specify whether patients provided:

    1) Consent to use their medical records/samples in research

    2) Consent to undergo the procedure

    3) Consent to take part in the study reported in this manuscript.

If the ethics committee waived the need for additional informed consent, please state this.

Thank you for your attention to these requests.

Reviewers' comments:

Reviewer's Responses to Questions

**Comments to the Author**

1. Is the manuscript technically sound, and do the data support the conclusions?

Reviewer #1: Yes

Reviewer #2: Yes

2. Has the statistical analysis been performed appropriately and rigorously? 

Reviewer #1: N/A

Reviewer #2: Yes

3. Have the authors made all data underlying the findings in their manuscript fully available?

Reviewer #1: Yes

Reviewer #2: Yes

4. Is the manuscript presented in an intelligible fashion and written in standard English?

Reviewer #1: Yes

Reviewer #2: Yes

5. Review Comments to the Author

Reviewer #1: 1. Detailed analysis of pazopanib and pneumothorax

2. Follow up data on what happened to pt was very helpful and need for pleurodesis is helpful for clinicians.

3. Timing of PTX and cavitation is instructive and helpful.

4. Case summary instructive if space needed, can omit

5.

Reviewer #2: 1. In Materials, p4 line 72, you indicated 19 patients evaluated, but all the findings are about 20 patients.

2. In Discussion, p 11 line 203, I think you 'observed' the difference between the behavior of central and subpleural lesions, rather than 'hypothesized' about it.

3. Use the terminology that the lesions or cavities were central or subpleural rather than the pneumothorax. The pneumothorax is always pleural in location.

4. Table 1 Cavitation of lesions, appears that 6 pts had ptx, 5/5 + 1/15. The text page 8 reads that 5/6 patients with cavitation had ptx and 0/14 patients without cavitation had ptx. Clarify this.

5. Did patients have multiple cavities? Both central and subpleural? Were recurrent ptx related to the same lesion or multiple lesions? Was the rupturing lesion always identified?

6. Figure 1 and 2 could probably be adequately done with 2 or 3 parts each instead of 3-5 parts. Or left out completely in favor of figures 3 and 4.

7. Add some description of time course from cavitation to ptx. Did cavities continually increase? Did some cavities increase but never rupture? Was there a size that made rupture likely? What was the minimum time from cavitation to rupture? Did cavities that didn't rupture persist or improve?

8. Although previous intervention was associated with more ptx, those numbers seem very small for any strong statement. Were the interventions spatially related to the cavitating lesions, i.e. was it the lesion previously intervened on or even the same lung, that went on to cavitate?

9. Were patients always symptomatic when ptx discovered?

6. PLOS authors have the option to publish the peer review history of their article (what does this mean?). If published, this will include your full peer review and any attached files.

Reviewer #1: No

Reviewer #2: **Yes: **Gregory W. Gladish, MD

---

## [Author Response · Author response to Decision Letter 0]

21 Jun 2021

Additional Editor Comments:

This is an interesting study that presents important and useful information, but it needs some clarification. Please recheck the numbers. You state that there were 32 patients treated, 4 were excluded due to extrapulmonary metastases, but that left 19 patients. What about the other 9? The manuscript focuses on 20 patients. Where did the other 1 come from? 

Response: Thank you for your comment. We apologize for these inconsistencies in numerical data. We have amended them accordingly (line 74).

More discussion should be placed on the extraordinarily high response rate, which is typically 5%, 

Response: The exact reasons why the number of pneumothorax cases was higher in this study are unclear, but we have presumed several reasons. 

1. Pre-existing cavity before pazopanib administration: In this study, there were patients who had pre-existing cavitation (Pt No. 2) and cystic changes in metastatic lesions after cessation of pazopanib (Pt No. 8), which acted as the pivot for the occurrence of pneumothorax. However, it is difficult to distinguish whether pazopanib facilitates augmentation of the pre-existing cavity or the natural course of metastatic lesions (line 246-254). 

2. Some patients with pneumothorax were asymptomatic during the diagnosis. The detection bias, which was caused by more frequent CT analysis, is the reason for the high rate of pneumothorax cases (line 272-274). 

3. The dose of pazopanib was related to the occurrence of pneumothorax. In this study, we found that the maximum dose of pazopanib (800 mg) was related to the incidence of pneumothorax (4/7 [800] vs. 1/13 [under 800], p=0.015, chi-square analysis) (line 197-199, 262-267).

4. Re-administration of pazopanib might be one of the reasons for the high rates. Until the acceptance of trabectedin or eribulin, the choice to treat the metastatic lesion had been limited in Japan, and we were reluctant to administer pazopanib again (line 240-245). 

and whether the "nonresponding" patient who had pneumothorax had a response that did not qualify for PR.

Response: Patients who achieved PR following pazopanib therapy occasionally experienced pneumothorax due to the degeneration of the tumor, resulting in an air cavity (4/9 [PR], 1/11 [SD or PD], p=0.069, chi-square analysis) (line 188-194). However, in thus study, we experienced cases of pneumothorax due to necrosis, which is caused by the natural progression of the tumor and is not related to the pazopanib response (pre-existing cavitation case, Pt no. 2). Pt no. 8 (44-year-old man, extraskeletal chondrosarcoma) experienced pneumothorax after cessation of pazopanib. This might be related to the natural characteristics of soft tissue sarcomas (line 262-267). 

Are Japanese patients more likely to respond to this therapy or simply more likely to develop pneumothorax than caucasians. Please address these questions as well as others raised in the review process.

Response: The fundamental study of pazopanib revealed no apparent racial differences in metabolism (Yanli Deng, et al. Bioavailability, metabolism and disposition of oral pazopanib in patients with advanced cancer, Xenobiotica, 43:5, 443-453, DOI: 10.3109/00498254.2012.734642). We considered the dose of the PALETTE study (800 mg) might be relatively higher for the small physique of our population. This might be one of the reasons for the high rate (264-271). 

Journal Requirements:

2. Please include the full name of the ethics committee that approved your study in the manuscript Methods.

Response: I have added the name of Ethics Committee in the Methods (line 114-117).

3. Please state whether the data were obtained retrospectively from medical records. If so, in the ethics statement in the manuscript and in the online submission form, please provide additional information about the patient records/samples used in your study, including: a) whether all data were fully anonymized before you accessed them; b) the date range (month and year) during which patients' medical records/samples were accessed; c) the source of the medical records/samples analyzed in this work (e.g. hospital, institution or medical center name).

Response: The medical records were retrospectively acquired. We have confirmed that all data were fully anonymized before the analysis. The data include medical records and samples analyzed in this study (line 76-77).

In addition, we note that you obtained consents from participants to take part in your study. In the Ethics Statement on the online submission form and the manuscript Methods , please clarify the context in which consent was obtained, and specify whether patients provided:

 1) Consent to use their medical records/samples in research

 2) Consent to undergo the procedure

 3) Consent to take part in the study reported in this manuscript.

Response: We have added these comments accordingly (line 114-117).

Reviewer #1: 1. Detailed analysis of pazopanib and pneumothorax

2. Follow up data on what happened to pt was very helpful and need for pleurodesis is helpful for clinicians.

3. Timing of PTX and cavitation is instructive and helpful.

4. Case summary instructive if space needed, can omit

Response: Thank you for your positive acknowledgment of our manuscript. 

Reviewer #2: 

In Materials, p4 line 72, you indicated 19 patients evaluated, but all the findings are about 20 patients.

Response: We apologize for this proofreading oversight. We have amended it accordingly (line 74). 

2. In Discussion, p 11 line 203, I think you 'observed' the difference between the behavior of central and subpleural lesions, rather than 'hypothesized' about it.

Response: You are correct. We have revised the description (line 220).

3. Use the terminology that the lesions or cavities were central or subpleural rather than the pneumothorax. The pneumothorax is always pleural in location.

Response: As you correctly mentioned, the pneumothorax is always pleural in location. We apologize for not being explicit earlier and have thus revised it to the exact location to avoid any misunderstanding.

4. Table 1 Cavitation of lesions, appears that 6 pts had ptx, 5/5 + 1/15. The text page 8 reads that 5/6 patients with cavitation had ptx and 0/14 patients without cavitation had ptx. Clarify this.

Response: Thank you for your comment. We apologize for this oversight. The correct description is 5/6, 0/14 (p<0.001); this has been amended in the revised manuscript (line 183-185, and Table).

5. Did patients have multiple cavities? Both central and subpleural? Were recurrent ptx related to the same lesion or multiple lesions? Was the rupturing lesion always identified?

Response: I have added the following information to the revised manuscript

Whether the patients had multiple cavities (line 141-144). 

Supplemental Table: whether the re-ruptured lesions were the same. In this study, we experienced a ruptured lesion alternatively on the left and right sides (Pt No. 2 and 3). 

The ruptured lesions were assumed by comparing consecutive CT images with the cavitated area. 

6. Figure 1 and 2 could probably be adequately done with 2 or 3 parts each instead of 3-5 parts. Or left out completely in favor of figures 3 and 4.

Response: Thank you for this valuable suggestion. I have reduced the number of figures in deference to the reviewer’s instructions (figure 2).

7. Add some description of time course from cavitation to ptx. 

Response: The detail of the time course has been depicted in annotations in the Supplemental Table. 

The patient (no. 2, 73-year-old woman, undifferentiated pleomorphic sarcoma) with bilateral multiple metastases with or without cavitation had been identified before administering pazopanib. For solid metastases, radiofrequency ablation was performed for bilateral lung metastases four times. Then, lobectomy of the left lower lobe was performed. Two months after surgery, pazopanib was administered (800 mg). At that time, there were many metastases with small cavitation (approximately 10 mm) and prominent cavitation in the left lung (S5, 26 mm, central lesion). After the initiation of pazopanib treatment, the patient’s condition was well controlled. However, the prominent cavity increased to 57 mm, though the other cavities did not increase. Subsequently, pneumothorax occurred. This pneumothorax was not related to the previously intervened area. After drainage and pleurodesis, the pneumothorax was maintained for 6 months. However, a second pneumothorax occurred in the same cavity.

Patient no. 4, 15-year-old, male, extraskeletal osteosarcoma: Bilateral multiple metastases without cavitation were identified. Two weeks after the initiation of pazopanib, approximately half of the metastatic lesions degenerated into several cavity. At the same time, the patient complained of chest pain; pneumothorax that occurred from the subpleural lesion of the right lung was detected. Two weeks after drainage, pazopanib was initiated. However, after 2 weeks, an asymptomatic small pneumothorax was noted in the subpleural lesion of the left lung. Subsequently, the continuation of pazopanib was considered difficult. However, repeated pneumothorax occurred on both sides. 

Patient no. 8 (44-year-old, male, extraskeletal chondrosarcoma): Under the administration of pazopanib, the lung metastases were controlled as stable disease. Six months after treatment, rapid growth and appearance of new lesions were noted. At the same time, cavitation of the subpleural lesion in the right lung appeared. Owing to the progressive form of the disease, pazopanib was discontinued. One week after cessation, pneumothorax occurred in the right lung from the cavitation area and was treated with drainage. After recovery from pneumothorax, re-initiation of pazopanib was difficult because of the patient’s condition. With an increase in the number of metastatic lesions, several lesions accompanied degenerated cysts. This indicated that these cavitations were not related to the administration of pneumothorax but rather to the natural course of the tumor.

Did cavities continually increase? Did some cavities increase but never rupture? Was there a size that made rupture likely? What was the minimum time from cavitation to rupture? Did cavities that didn't rupture persist or improve?

Response:

Indeed, some cavities that had increased and never ruptured existed. One patient (representative case, Figure 3) had several cavities in both the central and subpleural lesions. Pneumothorax occurred in the subpleural lesion, and the central lesion was stable. This implies that pneumothorax can easily occur in subpleural lesions (line 148 - 159).

The size of the cavity is an important independent factor. However, the location of the cavity and proximity to the pleural membrane are important. This is because pneumothorax occurs from the aperture to the ventral pleural membrane. We should be caution for the rapidly increasing central lesion, but pneumothorax from the subpleural lesion can frequently occur from a small lesion. It is difficult to define the cut-off value for size. 

Regarding the time from cavitation (at the detected day) to rupture, for the central type, it was 19 months (No. 1) and 3 months (No. 2), and for the subpleural type, it was 1 weeks (No. 3), the same day (No. 4), and 1 week (No. 8) (Supplemental table).

In this study, we did not experience any cavities that had shrunk. 

8. Although previous intervention was associated with more ptx, those numbers seem very small for any strong statement. Were the interventions spatially related to the cavitating lesions, i.e. was it the lesion previously intervened on or even the same lung, that went on to cavitate?

Response: As you correctly mentioned, the sample size was too small to provide such as a strong conclusion. Interventions to the lung were not directly related to the ruptured field (i.e., the cavitation was not related to anastomotic leakage). Indirect relationships via unknow factors (e.g., fibrosis around surgical site, burden for the remaining lung, stress by the invasive procedure and anesthesia) have been indicated (line 194-197).

9. Were patients always symptomatic when ptx discovered?

Response: I have expounded on this concern in the revised manuscript. The symptoms of pneumothorax were versatile, ranging from nearly asymptomatic to chest pain and dyspnea on exertion to severe dyspnea (line 129-132)

---

## [Editor Report · Decision Letter 1]

6 Jul 2021

Different patterns of pneumothorax in patients with soft tissue tumors treated with pazopanib: A case series analysis

PONE-D-21-10938R1

Dear Dr. Aiba,

We’re pleased to inform you that your manuscript has been judged scientifically suitable for publication and will be formally accepted for publication once it meets all outstanding technical requirements.

Kind regards,

Robert S Benjamin

Academic Editor

PLOS ONE

Additional Editor Comments (optional):

The authors have satisfactorily responded to the reviewers comments The manuscript is now acceptable for publication. I would suggest 2 further small revisions. In line 143, the close parenthesis should be moved to the end of the line; and in line 149, add the words "statistically significant" before the word difference.
---

## [Editor Report · Acceptance letter]

8 Jul 2021

PONE-D-21-10938R1 

Different patterns of pneumothorax in patients with soft tissue tumors treated with pazopanib: A case series analysis 

Dear Dr. Aiba:

I'm pleased to inform you that your manuscript has been deemed suitable for publication in PLOS ONE. Congratulations! Your manuscript is now with our production department. 

Kind regards, 

on behalf of

Professor Robert S Benjamin 

Academic Editor

PLOS ONE